# Glycation Interferes with the Expression of Sialyltransferases and Leads to Increased Polysialylation in Glioblastoma Cells

**DOI:** 10.3390/cells12232758

**Published:** 2023-12-02

**Authors:** Paola Schildhauer, Philipp Selke, Martin S. Staege, Anja Harder, Christian Scheller, Christian Strauss, Rüdiger Horstkorte, Maximilian Scheer, Sandra Leisz

**Affiliations:** 1Department of Neurosurgery, Medical Faculty, Martin Luther University Halle-Wittenberg, Ernst-Grube-Str. 40, 06120 Halle (Saale), Germany; paola.schildhauer@uk-halle.de (P.S.); maximilian.scheer@uk-halle.de (M.S.); 2Institute for Physiological Chemistry, Medical Faculty, Martin Luther University Halle-Wittenberg, 06114 Halle (Saale), Germany; 3Department of Surgical and Conservative Pediatrics and Adolescent Medicine, Medical Faculty, Martin Luther University Halle-Wittenberg, 06120 Halle (Saale), Germany; 4Institute of Neuropathology, University Medical Center, Johannes Gutenberg University Mainz, 55131 Mainz, Germany; 5CURE-NF Research Group, Medical Faculty, Martin Luther University Halle-Wittenberg, 06112 Halle (Saale), Germany

**Keywords:** glycation, glioblastoma, glioma, astrocytes, methylglyoxal, sialyltransferases, polysialylation

## Abstract

Glioblastoma (GBM) is a highly aggressive brain tumor that often utilizes aerobic glycolysis for energy production (Warburg effect), resulting in increased methylglyoxal (MGO) production. MGO, a reactive dicarbonyl compound, causes protein alterations and cellular dysfunction via glycation. In this study, we investigated the effect of glycation on sialylation, a common post-translational modification implicated in cancer. Our experiments using glioma cell lines, human astrocytes (hA), and primary glioma samples revealed different gene expressions of sialyltransferases among cells, highlighting the complexity of the system. Glycation has a differential effect on sialyltransferase expression, upregulating *ST8SIA4* in the LN229 and U251 cell lines and decreasing the expression in normal hA. Subsequently, polysialylation increased in the LN229 and U251 cell lines and decreased in hA. This increase in polysialylation could lead to a more aggressive phenotype due to its involvement in cancer hallmark processes such as immune evasion, resistance to apoptosis, and enhancing invasion. Our findings provide insights into the mechanisms underlying GBM aggressiveness and suggest that targeting glycation and sialylation could be a potential therapeutic strategy.

## 1. Introduction

Gliomas are the most common primary intracranial brain tumors and account for 80% of malignant brain tumors [1]. Based on the pathology of the neoplasia, the resulting survival gliomas are classified into four grades according to the World Health Organization (WHO) [2]. Tumors classified as grade 1 are low risk with slow growth and can be effectively treated with surgical removal. In contrast, grade 2 tumors are more invasive and prone to recurrence despite their low growth rate. Grade 3 tumors are typically considered malignant, characterized by abnormal cell growth and rapid cell division, while grade 4 gliomas are the most severe and have a poor prognosis, often leading to a fatal outcome [2,3]. The most common and aggressive glioma subtype is the glioblastoma (WHO grade 4). It is characterized by a high proliferation rate and diffuse infiltration into surrounding brain tissue, with a poor prognosis and a median overall survival of 15 months [4]. According to the latest WHO 2021 Classification of Central Nervous System tumors, only tumors lacking the isocitrate dehydrogenase (*IDH*) mutation are categorized as glioblastoma (GBM). In addition, one of the following five criteria needs to be present: microvascular proliferation or necrosis or telomerase reverse transcriptase (*TERT*) promotor mutation or epidermal growth factor receptor (*EGFR*) gene amplification or +7/–10 chromosome copy number changes. Besides *IDH*-wildtype GBM, the adult-type diffuse gliomas are further subdivided into astrocytoma *IDH*-mutant and oligodendroglioma (*IDH*-mutant and 1p/19-codeleted) [5,6].

In the diagnosis of gliomas, the genetic analysis and determination of molecular markers have become more valuable in recent years [7]. With these high throughput methods, four different subtypes of GBM (classic, neural, proneural, and mesenchymal) could be identified, which are characterized by distinct molecular profiles and clinical features, and they may respond differently to treatment. The classical subtype is characterized by high-level *EGFR* amplifications and 30% having *EGFR* mutations. The neural subtype displays a high expression of neuron markers such as neurofilament light chain (*NEFL*), gamma-aminobutyric acid receptor subunit alpha-1 (*GABRA1*), synaptotagmin-1 (*SYT1*), and solute carrier family 12 member 5 (*SLC12A5*). The mesenchymal subtype is strongly associated with a higher frequency of neurofibromatosis type 1 (*NF1)* mutations and upregulation of the tumor necrosis factor (*TNF*) and nuclear factor kappa-light-chain-enhancer of activated B cells (*NF-κB*). The proneural subtype is found to be closely linked to amplifications in platelet-derived growth factor receptor A (*PDGFRA*) and mutations in isocitrate dehydrogenase 1 *(IDH1*), which causes global hypermethylation and a phenotype called glioma-CpG island methylator phenotype (*G-CIMP*) [7,8,9].

GBM cells have the ability to switch metabolism to aerobic glycolysis (Warburg effect), which is also known as one of the hallmarks of cancer [10,11]. In the process of glycolysis, 0.4% of glucose is transformed into methylglyoxal (MGO), a potent glycation agent with pro-tumorigenic features [12,13]. MGO reacts nonenzymatically with proteins, lipids, and DNA, forming advanced glycation end products (AGEs). Post-translational modifications by MGO stimulate inflammation and oxidative stress, which can lead to DNA modifications and contribute to the development of cancer. Additionally, alterations in protein structure and functions are related to changes in the cell signaling pathways that promote cancer cell proliferation and survival [14,15,16]. Furthermore, AGEs can promote tumor growth by activating the receptor of AGEs (RAGE), as recent studies have shown that inducing RAGE expression in diabetic mice resulted in increased tumor growth in the GL261 glioma model via the upregulation of High Mobility Group Box 1 (*HMGB1*) [17]. In our previous study, we demonstrated that glycation with MGO increases the invasion of GBM cells, resulting in a more aggressive subtype [18]. Similar results were found in meningioma and neuroblastoma, where invasion was increased after glycation [19,20]. Interestingly, glycation increased polysialylated neural cell adhesion molecule (PolySia-NCAM) in neuroblastoma cells, where PolySia-NCAM is seen as an adverse prognostic marker [19]. In addition to this, glycation interfered with the expression of sialyltransferases (*STs*) in the meningioma cell line specifically [21].

Sialylation is the enzymatic addition of sialic acids to the terminal position of glycans, which form a part of the glycocalyx on the cell surface. This layer of complex sugar molecules plays a significant role in a variety of biological processes, such as cellular recognition, cell-to-cell adhesion, and cell signaling [22]. There are 20 human STs that use cytidine monophosphate-activated N-acetylneuraminic acid (CMP-Neu5Ac) as an active sugar donor. The STs are subdivided into four groups, depending on the glycosidic bond they form: β-galactoside-α-2,3-sialyltransferases (ST3GAL1-6), β-galactoside α-2,6-sialyltransferases (ST6GAL1-2), N-acetyl galactosamine α-2,6-sialyltransferases (ST6GALNAC1-6), and α-2,8-sialyltransferases (ST8SIA1-6) (Figure 1). The expression levels of these STs are tissue-specific and can also differ between cancer cells from the same origin [23].

A specific form of sialylation is the increased expression of the α-2,8-linked polymer, known as polysialic acid (polySia), which has been detected in several types of cancers and is associated with high-grade tumors [24]. The enzymes responsible for polysialylation are ST8SIA2 and ST8SIA4. PolySia is primarily bound to N-glycans of NCAM. PolySia-NCAM is predominantly expressed in neural tissue during embryonic development and has a significant role in neural development and synaptic plasticity [25]. However, it is associated with poor clinical outcomes and is re-expressed in several cancers, such as neuroblastoma, Wilms tumor, and lung carcinoma [26]. Additionally, higher-grade astrocytoma (WHO grades 3 and 4) express substantially more polySia compared to lower grades (WHO grades 1 and 2) [26]. Furthermore, in GBM, PolySia-NCAM has been considered a negative prognostic marker [27].

Aberrant sialylation is a widespread characteristic across many cancer types and has been recognized as a hallmark of cancer [27]. An increase in sialic acid, especially α-2,6 and α-2,3 linked sialylation, commonly known as hypersialylation has been closely associated with lung, breast, ovarian, and pancreatic cancer [28,29]. This increase in sialylation can be due to increased sialyltransferase expression, altered neuraminidase activity which cleaves sialic acids, or an increased availability of CMP-sialic acid [28]. Hypersialylation has been linked to the promotion of metastasis through integrin-mediated processes, such as the adhesion, migration, and signaling of metastatic cells [30]. In addition, the hypersialylation of receptors, such as Fas and tumor necrosis factor receptor 1 (TNFR1) death receptors, can protect against apoptosis and contribute to increased cancer cell survival [31]. Furthermore, sialylated glycans on cell surfaces play a role in immune system evasion by binding Siglecs on immune cells and inhibiting the cytotoxicity of natural killer cells and the activation of T-cells [32].

Sialylation has been identified as a critical player in the progression of cancer, including GBMs, where it is involved in migration, invasion, and immune evasion [33,34,35]. However, the impact of glycation on sialylation remains poorly understood. Here, we investigated the effect of glycation on sialyltransferase expression in GBM for the first time and compared the expression pattern of STs in glioma cell lines to that of normal human astrocytes and in nine primary gliomas.

## 2. Materials and Methods

### 2.1. Cell Lines and Cultivation

The human glioma cell lines U343, U251, and LN229 were cultured in RPMI 1640 (Gibco, Thermo Fisher Scientific, Waltham, MA, USA) supplemented with 1% penicillin-streptomycin (10,000 U/mL penicillin/10,000 µg/mL streptomycin) (Gibco, Thermo Fisher Scientific, Waltham, MA, USA) and 10% fetal bovine serum (FBS, Gibco, Thermo Fisher Scientific, Waltham, MA, USA) at 37 °C in a 5% CO_2_ incubator. The three cell lines, kindly provided by Jacqueline Kessler (Department of Radiotherapy, Martin Luther University Halle-Wittenberg, Halle (Saale), Germany), exhibit varying genetic origins and backgrounds (Appendix A). The U343 cell line’s origin was initially classified as a grade 3 anaplastic astrocytoma. Nevertheless, according to the latest WHO 2021 Classification of Central Nervous System tumors and recent literature, the U343 cell line can possibly be considered a grade 4 GBM cell line [5,36]. In this study, the U343 cell line is, however, consistently described as grade 3, as it was originally classified. Human astrocytes (hA), acquired from Sciencell Research Laboratories (Carlsbad, CA, USA), were used as a non-cancerous cell line. hA were cultured with astrocyte medium (Sciencell Research Laboratories, Carlsbad, CA, USA) according to the manufacturer’s recommendations. The plates for hA were precoated with poly-L-lysine solution (0.01%, EMD Millipore Corporation, Burlington, VT, USA).

### 2.2. Impedance Real-Time Cell Analysis

The impedance real-time cell analysis of the U343 cell proliferation was measured with the xCELLigence RTCA eSight (OmniLife Science, Bremen, Germany). For this, 5 × 10^3^ cells were seeded on E-Plates view 96-well (Agilent Technologies, Santa Clara, CA, USA). Different MGO concentrations (0.0, 0.3, 0.6 mmol/L) were added to the cells. Impedance measurement took place every 15 min and microscopic imaging every 90 min for 42 h. Cell index was calculated with eSight software version 1.1.2 (Cell Index = impedance_[tn]_ − impedance_[t0]_).

### 2.3. Immunoblotting

The treatment of cells with different MGO concentrations and protein extractions was carried out as previously described [18]. Subsequently, protein separation was performed using sodium dodecyl sulfate polyacrylamide gel electrophoresis (SDS-PAGE) with NuPAGE™ 4–12% gel, BIS-TRIS buffer, and 1.5mm Mini-Protein-Gels, along with the NuPAGE™ MES SDS Running Buffer (ThermoFisher Scientific, Waltham, MA, USA). Protein transfer from gel to membrane was performed using the iBlot 2 Dry Blotting System and iBlot™ 2 NC Regular Stacks (both Thermo Fisher Scientific, Waltham, MA, USA).

Ponceau S staining (0.1% Ponceau S, 3% trichloroacetic acid, and 3% sulfosalicylic acid) was used for rapid reversible detection of protein bands (Appendix A). The non-specific bindings were blocked with 5% skim milk powder (Carl Roth, Karlsruhe, Germany) in TRIS-buffered saline containing 0.1% Tween (TBS-T, Sigma-Aldrich, St. Louis, MO, USA), and the respective primary antibody was incubated overnight at 4 °C (Table 1). The primary antibody PolySia was kindly donated by Professor Rita Gerardy-Schahn, Hannover Medical School, Hannover, Germany). Secondary antibodies were subsequently added to the membranes for 60 min at room temperature. Glyceraldehyde 3-phosphate dehydrogenase (GAPDH) protein levels were used as a loading control. To visualize the bound antibodies, SuperSignal™ West Pico PLUS Chemiluminescent Substrate (Thermo Fisher Scientific, Waltham, MA, USA) was used and the signals were captured using a CCD camera (ImageQuant LAS4000, GE Healthcare, Freiburg, Germany). The band intensity was quantified using ImageQuantTL v2005 software (GE Healthcare, Freiburg, Germany) and normalized to the corresponding GAPDH bands.

### 2.4. Immunohistochemistry (IHC)

Formalin-fixed paraffin-embedded (FFPE) GBM tissue sections were deparaffinized, heated to 95 °C for 6 min, and cooled under running water. The tissue samples were subsequently treated with a solution of 3% H_2_O_2_ (Carl Roth, Karlsruhe, Germany) to block endogenous peroxidase activity, followed by two washes using PBS with Tween 20 (PBST). Biotin blocking was performed using the Avidin/Biotin Blocking Kit (Vector Laboratories, Newark, CA, USA) according to manufacturer’s instructions. For immunohistochemistry with an anti-polySia recombinant monoclonal antibody (clon 735, Table 1) and anti-carboxymethyl lysine antibody (CML26, ab125145, Abcam, Cambridge, UK) slides were incubated at a concentration of 1:200 (polySia antibody) and of 1:50 (carboxymethyl lysine antibody) overnight and washed twice with PBST thereafter. Incubation with a secondary antibody followed at room temperature using the Vectastain anti-mouse IgG biotinylated antibody (Vector Laboratories, Newark, CA, USA) according to manufacturer’s instructions. After washing twice with PBST incubation with DAB Quantolab (Epredia, Kalamazoo, MI, USA) was performed using 1000 µL substrate and 20 µL chromogen (Vectastain Elite ABC Kit, Vector Laboratories, Newark, CA, USA). Slides were then washed with aqua dest and incubated with hematoxylin 7211 (Epredia, Kalamazoo, MI, USA) for 1 min, again washed with water for 5 min, and dried. Finally, the slides were mounted with CytoSeal medium. Microscopy was performed with a DM 2500 LED microscope (Leica Microsystems, Wetzlar, Germany) and scanned with a NanoZoomer 2.OHT (Hamamatsu Photonics, Shizuoka, Japan). Stainings were evaluated visually as well as using NDP.view2 software (Hamamatsu Photonics, Shizuoka, Japan) by a neuropathologist. A FFPE-embedded cell pellet (cytoblock) of LN229 cells was used as positive control and for optimization.

### 2.5. mRNA Isolation and qPCR

mRNA isolation and cDNA transcription were conducted according to the previously described methods [18]. Quantitative polymerase chain reaction (qPCR) was performed using Platinum^®^ SYBR^®^ Green qPCR SuperMix-UDG (Invitrogen, Thermo Fisher Scientific, Waltham, MA, USA) 0.5 µL of the respective reverse and forward primers (Table 2), and 1 µL of the cDNA. The qPCR was carried out using the Rotor-Gene Q machine (Qiagen, Hilden, Germany). The 2^-ΔΔCT^ method was used for the analysis. The mean value of the mRNA level of the replicates from the untreated cells was set to 1. Only for the heat map in Figure 2, the 2^-ΔCT^ values were used for better comparability.

### 2.6. Glioma Primary Cultures

Tumor material was collected using a cavitron ultrasonic surgical aspirator (CUSA) during microsurgical removal of GBM. The tumor material was washed with PBS (Thermo Fisher Scientific, Waltham, MA, USA) three to four times and then incubated with a digestive solution composed of 2.5% trypsin (Thermo Fisher Scientific, Waltham, MA, USA) and 250 µg/mL DNase (AppliChem GmbH, Darmstadt, Germany) in Hanks’ Balanced Salt Solution (HBSS without Ca^2+^ and Mg^2+^, Thermo Fisher Scientific, Waltham, MA, USA) for five to ten minutes at 37 °C. Tissue digestion was stopped by adding 10 mL of Dulbecco’s Modified Eagle’s Medium (DMEM, Thermo Fisher Scientific, Waltham, MA, USA) supplemented with 1% penicillin-streptomycin and 10% FBS, followed by centrifugation and resuspension of the cell pellet in 10 mL of complete DMEM. The resulting cell suspension was seeded into 75 cm^2^ tissue culture flasks (Sarstedt, Nümbrecht, Germany) and incubated at 37 °C and 5% CO_2_ under a humidified atmosphere. The primary cultures were tested for hepatitis B virus, hepatitis C, and human immunodeficiency virus, and only passages below five have been used.

### 2.7. RNAseq Analysis

Nine primary cultures were randomly selected for RNA sequencing (RNAseq) analysis. RNA isolation was carried out as described in Section 2.3. RNAseq data was generated from 1 µg RNA by Novogene (Cambridge, UK) using the Illumina Novaseq6000 system. A further data procedure was conducted, essentially as described [37], using the Galaxy server (https://usegalaxy.eu/ (accessed on 24 November 2022)). Reads were mapped against the human reference genome HG38 using HISAT2, and quantification was performed using htseq-count.

### 2.8. Statistical Analysis

Excel Software 2016 (Microsoft Corporation, Redmond, WA, USA) was used for all analysis, and visualizations were performed using GraphPad Prism 4.9.1 (GraphPad Software Inc., San Diego, CA, USA). To evaluate statistical significance, a paired student´s *t*-test was performed, respectively, for all cell lines. The average means are depicted in the figures with error bars that represent standard deviations (SD). For each experiment, at least three biological replicas were conducted. The expression levels of STs in central nervous system (CNS) tumors and normal tissue were plotted and compared using TNMplot (differential gene expression analysis in tumor, normal, and metastatic tissues database (https://tnmplot.com/analysis/ (accessed on 19 March 2023)) [38]. Comparisons were performed using the Mann–Whitney U test with a statistical significance threshold set at *p* value < 0.01.

### 2.9. Study Approval

The study was approved by the ethics committee of the Medical Faculty, Martin Luther University Halle-Wittenberg (approval number 2021-129) and conducted according to the Declaration of Helsinki. Written informed consent was obtained from all patients.

## 3. Results

### 3.1. mRNA Levels of Sialyltransferases in Glioma Cell Lines Compared to hA

Since STs are heterogeneously expressed in different cells and have pro-tumorigenic characteristics, we analyzed the expression pattern with qPCR in the malignant cell lines LN229, U251, and U343 and compared them to the normal hA. The cell lines showed a differential gene expression pattern of STs (Figure 2).

*ST3GAL1* exhibited higher mRNA level in LN229 (9.73 × 10^−4^ ± 1.89 × 10^−4^, *p* = 0.026) and U251 cells (1.17 × 10^−3^ ± 3.14 × 10^−4^, *p* = 0.022) compared to hA. In contrast, *ST3GAL2* showed markedly lower mRNA levels in LN229 cells (2.78 × 10^−4^ ± 1.44 × 10^−4^, *p* = 0.009) and U343 cells (5.91 × 10^−5^ ± 9.83 × 10^−6^, *p* = 0.015). Similarly, *ST3GAL3* also showed lower mRNA levels in LN229 cells (2.17 × 10^−4^ ± 1.01 × 10^−4^, *p* = 0.006. Furthermore, *ST3GAL4* exhibited lower mRNA levels in U343 cells (4.02 × 10^−4^ ± 6.79 × 10^−5^, *p* = 0.004) compared to hA. Interestingly, in LN229 cells (1.96 × 10^−5^ ± 6.17 × 10^−6^, *p* = 0.022), *ST3GAL5* showed significantly higher mRNA levels than in hA. However, U251 cells (1.29 × 10^−6^ ± 6.01 × 10^−4^, *p* = 0.025) and U343 cells (8.99 × 10^−7^ ± 3.86 × 10^−7^, *p* = 0.018) cells displayed lower mRNA levels of *ST3GAL5* compared to hA. Lastly, *ST3GAL6* demonstrated lower mRNA levels in LN229 cells (2.61 × 10^−5^ ± 6.35 × 10^−6^, *p* < 0.001) and U343 cells (3.36 × 10^−5^ ± 1.32 × 10^−5^, *p* < 0.001) cells compared to hA (Appendix A).

In the ST6GAL family, *ST6GAL1* showed higher mRNA levels in the U343 cells (5.85 × 10^−4^ ± 1.57 × 10^−4^, *p* = 0.019) compared to hA. Interestingly, *ST6GAL2* was only expressed in hA (Appendix A).

In all three malignant cell lines, *ST6GALNAC1* was detected in extremely low amounts. No expression was detected in hA. *ST6GALNAC2* showed higher mRNA levels in the LN229 cells (1.71 × 10^−3^ ± 3.73 × 10^−4^, *p* = 0.004) but showed lower expression in the U343 cells (1.59 × 10^−5^ ± 6.46 × 10^−6^, *p* = 0.016) in comparison to hA. *ST6GALNAC3* also showed remarkably low mRNA levels in the U343 cells (4.41 × 10^−7^ ± 2.63 × 10^−7^, *p* = 0.004). Moreover, *ST6GALNAC5* showed a higher mRNA level in the U251 cells (5.10 × 10^−4^ ± 1.80 × 10^−4^, *p* = 0.044), while *ST6GALNAC6* showed lower in the LN229 cells (1.14 × 10^−4^ ± 2.45 × 10^−5^, *p* = 0.036) (Appendix A).

Lastly, *ST8SIA2* was only detected in the U251 cells and hA, where U251 cells demonstrated lower expression (2.57 × 10^−4^ ± 1.15 × 10^−4^, *p* = 0.029). *ST8SIA3* was only detected in hA. *ST8SIA4*, which was not detected in the U343 cells, showed a lower mRNA level in the U251 cells (5.63 × 10^−5^ ± 1.75 × 10^−5^, *p* = 0.004) compared to hA. *ST8SIA5* showed lower mRNA levels in the U251 cells (2.21 × 10^−5^ ± 6.26 × 10^−6^, *p* = 0.039) and U343 cells (1.67 × 10^−5^ ± 3.212 × 10^−6^, *p* = 0.038) in comparison with hA. *ST8SIA6* revealed lower mRNA levels in all three malignant cell lines (LN229 cells: 6.99 × 10^−5^ ± 1.35 × 10^−5^, *p* = 0.001; U251 cells: 2.13 × 10^−4^ ± 1.20 × 10^−5^, *p* = 0.005; U343 cells: 2.19 × 10^−4^ ± 7.65 × 10^−5^, *p* = 0.005) (Appendix A).

### 3.2. Sialyltransferase Expression in Nine Primary Glioma Samples

The sialyltransferase mRNA of nine primary gliomas was analyzed using RNA sequencing analyses. The sample set included four cases of glioblastoma (WHO grade 4), two cases each of oligodendroglioma and astrocytoma (both WHO grade 2), and one case of ganglioglioma (WHO grade 1). The patients included had a mean age of 54 years (32 to 74 years). The baseline data of the patients are shown in Table 3.

The RNAseq data analysis using the HISAT aligner revealed a distinct differential gene expression pattern of STs among the samples. However, no significant differences in gene expression were observed between the WHO groups, indicating similar expression profiles within those groups (Figure 3).

Remarkably, the genes *ST6GALNAC4* and *ST6GALNAC6* displayed consistent reads across all analyzed samples, suggesting higher gene expression levels. Similarly, *ST3GAL2* and *ST3GAL4* exhibited elevated read levels in all samples compared. Intriguingly, sample number one exhibited a higher read count for the *ST3GAL1* gene compared to the other samples, implying a potential overexpression of this gene in that particular sample.

Further examination of the ST3GAL family revealed that *ST3GAL3*, *ST3GAL5*, and *ST3GAL6* exhibited relatively low read counts in all samples. In contrast, *ST6GAL1* showed high reads in samples four and five, while *ST6GALNAC2* exhibited consistently low read counts across all samples. The remaining members of the *ST6GALNAC* family showed low read counts and thus probably indicated low expression, except for *ST6GALNAC5*, which exhibited a higher fold increase in read count in sample one.

Among the *ST8SIA* family, most members displayed low read counts in all samples. However, *ST8SIA2* exhibited a higher read count in sample number eight, while *ST8SIA5* showed elevated read numbers in sample number one.

### 3.3. Expression of Sialyltransferase Genes in CNS Tumors When Comparing Paired Data from Normal and Tumor Gene Arrays

When comparing the mRNA level of the ST3GAL family in public microarray data from CNS tumors and normal tissue, *ST3GAL1, 2, 4,* and *5* were found to have lower expressions in the tumor samples compared to normal tissue. *ST3GAL6*, on the other hand, did not show any significant difference in gene expression between the two tissue types (Appendix A). Regarding the ST6GAL family, *ST6GAL1* showed higher expression levels in tumor tissue compared to normal tissue (Appendix A). Among the ST6GALNAC family, *ST6GALNAC2* and *4* exhibited higher expression in tumor samples than in normal tissues, while *ST6GALNAC5* had lower expression in tumor tissues than in normal tissues (Appendix A). Lastly, in the ST8SIA family, *ST8SIA1* and *4* had higher expressions in tumor tissues compared to normal tissues. The expressions of *ST8SIA2, 3,* and *5* were lower in the tumor samples than in normal tissues (Appendix A).

### 3.4. MGO-Dependent Cell Growth

In comparison to the untreated cells (Appendix A), a lower concentration of MGO (0.3 mmol/L) stimulated cell proliferation (Appendix A, Figure 4), while a higher concentration of MGO treatment (0.6 mmol/L) led to reduced cell proliferation (Appendix A, Figure 4), as shown for the U343. Due to this observation and previous work [18], the results of the following mRNA analyses were shown with 0.3 mmol/L MGO.

### 3.5. Cell Line Specific Alterations of Sialyltransferase mRNA Levels after MGO Treatment

To determine the suitability of MGO as a glycation agent, we exposed cells to various concentrations of MGO and utilized an anti-carboxymethyl lysine antibody to detect glycation, which has been published previously [18]. Our findings indicate that MGO treatment led to an elevation in glycation levels in a concentration-dependent manner. (Figure 5). The Western blots were quantified and revealed an increase in glycation after treatment with MGO at concentrations as low as 0.3 mmol/L for LN229 cells, U251 cells, and hA (Appendix A). Notably, in the case of U343 cells, the glycation signal was significantly increased after treatment with 0.6 mmol/L MGO.

Furthermore, we analyzed the effect of glycation on the mRNA level of the STs in the different cell lines after treatment with 0.3 mmol/L MGO. Glycation led to cell line-specific alterations in STs. Overall, the upregulation of STs was observed in LN229 cells after treatment with 0.3 mmol/L MGO and downregulation in the remaining cell lines (Appendix A). ST families are presented in detail in Figure 6, Figure 7, Figure 8 and Figure 9. The mRNA levels are represented as the fold change of untreated cells to treated cells.

### 3.6. Differential Regulation of ST3GAL Family after MGO Treatment

In cell line LN229, the mRNA level of *ST3GAL3* (1.294 ± 0.114, *p* = 0.035) was increased in contrast to the untreated cells (Figure 6A). Glycation decreased the mRNA level of *ST3GAL1* (0.280 ± 0.188, *p* = 0.040) and *ST3GAL5* (0.309 ± 0.044, *p* = 0.047) in U251 cells (Figure 6B). In cell line U343, the mRNA levels of *ST3GAL2* (0.507 ± 0.137, *p* = 0.021) and *ST3GAL4* (0.594 ± 0.086, *p* = 0.036) were downregulated (Figure 6C). Furthermore, the mRNA levels of *ST3GAL3* (0.569 ± 0.161, *p* = 0.045), *ST3GAL4* (0.474 ± 0.169, *p* = 0.028), *ST3GAL5* (0.285 ± 0.187, *p* = 0.029), and *ST3GAL6* (0.675 ± 0.067, *p* = 0.039) were decreased in hA (Figure 6D).

### 3.7. Differential Regulation of ST6GAL Family after MGO Treatment

In the ST6GAL family, *ST6GAL1* was not altered after MGO treatment in LN229 cells (Figure 7A) or in the U343 cells (Figure 7C) but downregulated in cell line U251 (0.270 ± 0.230, *p* = 0.039) and hA (0.396 ± 0.084, *p* = 0.023) (Figure 7B,D). The mRNA level of *ST6GAL2*, which was only detected in hA, was decreased after MGO treatment (0.103 ± 0.017, *p* = 0.008) (Figure 7D).

### 3.8. Differential Regulation of ST6GALNAC Family after MGO Treatment

Similar effects of glycation in the ST6GALNAC family were observed: upregulation of STs in LN229 cells and downregulation in U251 cells, U343 cells, and hA (Figure 8). In cell line LN229, *ST6GALNAC5* (1.383 ± 0.031, *p* = 0.049) and *ST6GALNAC6* (1.708 ± 0.150, *p* = 0.023) were upregulated after MGO treatment (Figure 8A). The downregulation of *ST6GALNAC6* (0.438 ± 0.204, *p* = 0.039) was observed in cell line U251 (Figure 8B). Moreover, the mRNA levels of *ST6GALNAC3* (0.446 ± 0.101, *p* = 0.017) and *ST6GALNAC6* (0.590 ± 0.052, *p* = 0.009) were decreased in cell line U343 (Figure 8C). In hA, glycation decreased the mRNA level of *ST6GALNAC3* (0.431 ± 0.107, *p* = 0.075) (Figure 8D).

### 3.9. Differential Regulation of ST8SIA Family after MGO Treatment

Finally, we observed the effect of glycation on the ST8SIA1-6 family (Figure 9). *ST8SIA2*, which was only detected in U251 cells and hA, was downregulated in U251 cells after glycation (0.208 ± 1.39, *p* = 0.020) (Figure 9B). The mRNA level of *ST8SIA3* was not affected after MGO treatment. The mRNA level of *ST8SIA4* was increased in LN229 cells (1.457 ± 0.166, *p* = 0.022) and U251 cells (1.457 ± 0.166, *p* = 0.021) (Figure 9A,B). In contrast, the mRNA level of *ST8SIA4* was downregulated in hA (0.527 ± 0.059, *p* < 0.001) after MGO treatment (Figure 9D). Moreover, the *ST8SIA5* mRNA level was decreased in cell line U343 (0.501 ± 0.028, *p* = 0.0209) (Figure 9C). *ST8SIA6* level decreased in U251 cells (0.289 ± 0.146, *p* = 0.038) and in hA (0.526 ± 0.045, *p* = 0.014) (Figure 9B,D).

### 3.10. Glycation Leads to Increased Polysialylation in GBM Cell Lines

To examine the impact of glycation on polysialylation, glioma cell lines were subjected to different concentrations of MGO, and the extracted proteins were separated using gel electrophoresis. PolySia antibody was used to detect polysialylation. Polysialylation increased in LN229 cells and U251 cells after MGO treatment. In the LN229 cell line, 0.3 mmol/L MGO increased polysialylation by 14.82% ± 5.82%, *p* = 0.023, and 1 mmol/L MGO increased polysialylation by 15.23% ± 3.68%, *p* = 0.004 (Figure 10A). In cell line U251, polysialylation increased in a concentration-dependent manner, and 0.3 mmol/L increased polysialylation by 13.11% ± 3.74%, *p* = 0.008, 0.6 mmol/L by 17.38% ± 1.42%, *p* < 0.001 and 1 mmol/L by 28.53% ± 14.36%, *p* = 0.048 (Figure 10B). In contrast to this, glycation decreased polysialylation in hA. 0.3 mmol/L MGO decreased polysialylation by 22.91% ± 1.65%, *p* < 0.001, and 1 mmol/L MGO decreased polysialylation by 14.86% ± 7.13, *p* = 0.043 (Figure 10C). Polysialylation was not detected in the U343 cell line and was not induced after MGO treatment (Appendix A).

### 3.11. IHC Analysis of Glycation and Polysialylation in Primary GBM

Moreover, IHC analysis was performed to investigate the localization and abundance of glycation and PolySia in primary GBM. The baseline data of the primary GBM samples are shown in the supplement (Appendix A). The results revealed a presence of polysialylation and glycation, which was variable concerning different tumor areas and also concerning intensity. However, protein modifications, glycation, and polysialylation were observed in the same regions of tumor tissue, as exemplarily shown (Figure 11A). Notably, at elevated concentrations of carboxymethyl lysine, polySia was increased in the same region (Figure 11B).

## 4. Discussion

Aberrant sialylation from dysregulated STs has been shown to play a key role in tumor progression and has gained importance in cancer research [39]. Sialic acids, terminally present on N-linked glycans of glycoproteins and α_3_β_1_ integrins in the glycocalyx, have been demonstrated to promote migration, invasion, chemotherapy resistance, and immune evasion in GBM [40,41]. In recent years, the influence of glycation on cancer development has gained momentum. Many studies could demonstrate the pro-tumorigenic effects of MGO, a reactive dicarbonyl, promoting migration, invasion, tumor growth, and evading apoptosis in various cancer types [12,13,15,42]. However, the effect of glycation on sialylation has been explored very little so far. For the first time, we analyzed the effect of glycation on the mRNA level of the STs in human glioma cell lines and human astrocytes.

We observed cell line-specific effects of glycation on sialyltransferase mRNA levels. Interestingly, we noted a general upregulation of STs in LN229 cells (GBM, WHO grade 4) and downregulation in U251 cells (GBM, WHO grade 4), U343 cells (glioma, WHO grade 3), and hA. Similar results were reported by Selke et al. who identified cell line-specific alterations of sialyltransferase expression after glycation in meningioma cell lines. Glycation increased sialyltransferase expression in BEN-MEN-1 benign meningioma cells (WHO grade 1) and decreased expression in IOMM-Lee malignant meningioma cells (WHO grade 3) [21].

Sialyltransferase expression is known to be regulated at the transcriptional level by oncogenes, transcription factors, miRNAs, long non-coding RNAs, gene amplification, hormones, and natural compounds [43]. It appears that the regulation of expression differs between STs, which may explain the inconsistent effect of glycation on sialyltransferase expression. For example, hypermethylation occurs within the CpG islands of the P3 promoter in GBM, resulting in a decreased expression of *ST6GAL1* [44,45]. Other authors have described the upregulation of *ST6GAL1* as a result of gene amplification or through oncogenic signaling pathways [45]. Bork et al. suggest that the availability of substrate from activated sialic acid regulates the expression of STs [46,47]. When less substrate is available, more enzymes are expressed to ensure proper sialylation, with the exception of ST8SIA4, which does not appear to be regulated by the donor substrate. This suggests that polysialylation of NCAM is regulated independently of general sialylation [47]. Moreover, there is strong evidence for post-translational regulation by mechanisms such as oligomerization, subcellular localization, and proteolytic processing, which could be affected by glycation [43]. One plausible mechanism through which glycation could impact the expression of STs is by interacting with histones. MGO exhibits a preference for reacting with arginine and, to a lesser extent, lysine residues in proteins, a characteristic that is notably present in histones. Emerging evidence indicates that the glycation of histones by MGO influences the epigenetics of the DNA, thereby causing transcriptional changes [48].

Of additional interest, we examined the STs expression pattern in the glioma cell lines in comparison to hA. Furthermore, the expression patterns of nine primary glioma samples obtained from patients with different WHO grades were examined. We reported a differential gene expression pattern of STs between the cell lines. The variability of expression could be due to the different genetic background of the cells or their different degree of differentiation.

Interestingly, in the malignant cell lines sialyltransferase mRNA levels were lower in comparison to hA with only a few exceptions. The RNA seq analysis of the STs performed on patient-derived glioma primary cultures confirmed this variability. However, no association between the sialyltransferase pattern and WHO grade was observed.

*ST3GAL1* was the only sialyltransferase that was expressed higher in the GBM cell lines compared to hA. In the primary samples, *ST3GAL1* was upregulated in one sample with a WHO grade 4 GBM.

The overexpression of ST3GAL1 has been linked to a negative prognosis in various cancers, such as colorectal, breast, ovarian, and bladder cancer, and gliomas [23,49]. Chong et al. discovered that the activation of ST3GAL1 through the TGFβ signaling pathway plays a role in enhancing invasiveness and tumorigenicity in mesenchymal-subtype GBM. Interestingly, the authors observed a statistically significant association between low *ST3GAL1* expression in low-grade gliomas and *IDH1* mutations. This study also showed that the increased expression of *ST3GAL1* characterizes an invasive subset of GBM cells that are capable of self-renewal and suppressing its function has been shown to prolong survival in mouse models. Additionally, the study demonstrated that the transcriptomic program associated with *ST3GAL1* is associated with a poor prognosis in glioma patients and results in higher tumor grades in the mesenchymal subtype [34].

The expressions of ST3GAL3, 4, and 6 are known to be higher in high-grade gliomas compared to low-grade gliomas. These enzymes are involved in the production of terminal sialylated lewis antigens (SLex), which are known to promote tumor metastasis [50]. Analysis of the primary glioma data showed that members of the ST3GAL family, specifically *ST3GAL2* and *ST3GAL4*, exhibited higher levels of expression [41].

ST6GAL1 is one of the most studied STs, contributing to the addition of alpha 2,6-linked sialic acid, and is frequently overexpressed in malignant cells. High expression correlates with higher tumor grade and poor prognosis in a variety of cancers [51,52]. In our study, *ST6GAL1* was expressed at higher levels in half of our primary samples and at higher levels in cell line U343. Interestingly, ST6GAL1 has been shown to protect tumor cells from hypoxic stress by increasing the expression of hypoxia-inducible factor alpha (HIF-1α) [53]. Nonetheless, the role of α-2,6 sialylation mediated by ST6GAL1 in GBM is unclear. Gc et al. reported that GBM cells with high α2,6 sialylation due to ST6GAL1 expression exhibited increased growth and self-renewal capacity and decreased survival in mice when orthotopically injected [54]. However, other studies showed that GBM cells overexpressing ST6GAL1 lose in vitro invasion and were not able to induce intracranial tumors, whereas ST3GAL3-mediated 2,3-linked sialic acid transfection increased invasive potential [55,56].

Of the ST6GALNAC family, we observed that *ST6GALNAC2* was higher expressed in LN229 cells and *ST6GALNAC5* in U251 cells than in hA. In colorectal carcinomas, the expression of *ST6GALNAC2* was linked to increased metastasis. However, in breast cancer, Murugaesu et al. described it as a metastasis suppressor. This contrast underscores the complexity and unique roles of sialyltransferases in different tumor types [57,58]. Interestingly, all primary gliomas showed high read numbers of *ST6GALNAC4* and *ST6GALNAC6* compared to the other STs. According to Man et al., the presence of ST6GALNAC4 results in abnormal glycosylation within hepatocellular carcinoma, consequently increasing proliferation, migration, and invasion via the activation of the TGF-β pathway in in vitro and in vivo analysis. Furthermore, ST6GALNAC4 contributes to the immune evasion strategy of hepatic carcinoma through its involvement in the T antigen-galectin3+ tumor-associated macrophages axis [59]. In human leukemia cells, the presence of ST6GALNAC4 leads to the generation of a disialyl-T glycan. This glycan functions as an immunosuppressive signal by interacting with macrophage Siglec-7, consequently impeding the clearance of cancer cells [60].

Furthermore, no expressions of *ST6GAL2* and *ST8SIA3* were detected in the malignant cell lines. They also showed very low expression in the nine primary glioma samples. *ST8SIA3* is expressed at very low levels compared to intracranial tumors according to the TNMplot data, although ST8SIA3 has been shown to promote survival, proliferation, and migration of glioblastoma cells based on ST8SIA3 knockdown experiments in vitro [61].

Although ST8SIA1, also known as GD3 synthase, is thought to be one of the key drivers of malignancy in GBM, we observed remarkably low expression of *ST8SIA1* in our patient data. ST8SIA1 expression positively correlates with an increasing grade of astrocytoma and is highly expressed in GBM according to Ohkawa et al. In a murine model of glioma, ST8SIA-deficient mice exhibited reduced glioma progression, lower-grade pathology, and longer overall survival [35].

The ST8SIA family was expressed low in our primary data set of gliomas, with the exception of *ST8SIA5* in sample one and *ST8SIA2* showing slightly higher read numbers. However, according to Dong et al. and the TMNplot data, *ST8SIA4* was significantly upregulated in GBM compared to non-neoplastic brain tissue [62].

Interestingly, U251 cells and hA expressed both sialyltransferases responsible for polysialylation: *ST8SIA2* and *ST8SIA4*. LN229 cells expressed only *ST8SIA4*, and the glioma cell line U343 expressed none. The expression of *ST8SIA2* was downregulated in the U251, whereas the expression of *ST8SIA4* increased in LN229 cells and U251 cells and decreased in hA. We observed an increase in polysialylation in LN229 cells and U251 cells following glycation and a decrease in hA, in line with the expression of *ST8SIA4*. In the IHC analysis of primary GBM, the co-occurrence of polysialylation and glycation was observed within overlapping regions, thereby reinforcing the notion that glycation could potentially contribute to increased polysialylation in gliomas and GBM.

Remarkably, glycation increased the invasion of LN229 cells and U251 cells in our previous study [18]. An increase in invasiveness, following glycation could potentially be explained by an increase in polysialylation. Scheer et al. observed an increase in the polysialylation of NCAM, while NCAM expression itself was not altered, and a subsequent increased invasion in the neuroblastoma cell line Kelly following MGO treatment [19]. The role of PolySia-NCAM in invasion has been established by Suzuki et al., who demonstrated that C6 glioma cells became highly invasive in a mouse model after increasing polysialylation via the transfection of *ST8SIA2* or *ST8SIA4*, while mock-transfected C6 glioma cells exhibited minimal invasion when inoculated into the brain [26]. These findings provide compelling evidence that polysialylation promotes tumor cell migration and invasion. PolySia-NCAM has been found to be more prevalent in cells of diffuse astrocytoma, which are known to spread extensively compared to other gliomas [26]. Furthermore, the presence of PolySia-NCAM has been identified as a negative predictor for prognosis in GBM and has been demonstrated to regulate oligodendrocyte transcription factor (OLIG2), a key factor oligodendrocyte differentiation involved in the tumor initiation, proliferation, and phenotypic plasticity [27,63,64].

In a study conducted by Rosa et al., it was observed that hypoxia results in an upregulation of *ST8SIA2* and *ST8SIA4* expression, as well as an increase in polySia levels. The study demonstrated that this rise in polySia levels plays a critical role in cell migration under low oxygen concentrations. The authors also demonstrated that GBM stem-like cells depend on PolySia-NCAM to maintain their undifferentiated state. Moreover, high levels of PolySia-NCAM were found in HIF-1α positive regions of GBM tissues, providing support for the theory that the hypoxic microenvironment could induce polysialylation and have both physiological and pathological implications in vivo [65].

In concurrence, hypoxia has been observed to rapidly induce AGE formation along with activation of receptors for AGE (RAGE) signaling [66]. AGE-RAGE interaction activates NADPH oxidases, generating reactive oxygen species (ROS) and resulting in a hypoxic condition. Moreover, HIF-1α stabilization promotes the shift of tumor metabolism towards aerobic glycolysis to adapt to hypoxic conditions, leading to the accumulation of dicarbonyls and other AGEs. This is achieved by upregulating glycolysis-related genes and glucose transporter (GLUT) 1. Consequently, tumor cells with heightened glucose uptake and hypoxic environments contribute to glycation, potentially leading to the upregulation of STs and increased polysialylation [67,68].

PolySia has been found to play a pivotal role in tumor cell migration and invasion, as well as promoting stemness and potentially contributing to immune evasion mechanisms [24,25,69]. Therefore, inhibiting polysialylation could be a promising therapeutic strategy against GBM. As demonstrated by Rosa et al., the use of the sialic acid analog fluorinated-N-acetylneuraminic acid (F-NANA) effectively lowered polySia levels in GBM cells, resulting in the inhibition of cell migration [65]. In addition, Putthisen et al. could demonstrate that suppression of sialylation by sialyltransferase inhibitor 3Fax-Peracetyl Neu5Ac enhanced the sensitivity of GBM cells to chemotherapeutic drugs [70].

## 5. Conclusions

In our study, we demonstrated that glioma cell lines and primary glioma cells have a differential expression pattern of STs. In addition, MGO treatment led to differential effects on the expression pattern in glioma cell lines, indicating that the regulatory mechanisms of the cells are distinct. Notably, *ST8SIA4* was upregulated in two cell lines following glycation, leading to increased polysialylation, which may contribute to enhanced invasion and a potentially more aggressive phenotype. Further research is needed to explore the mechanism between glycation and sialylation. Additionally, further investigation should be conducted to better understand the role of sialylation and polysialylation in GBM growth and invasion, as well as their potential as biomarkers for diagnosis and treatment.

## 6. Limitations

The scope of our study is constrained by the use of a few cell lines to investigate the effects of glycation, and these cell lines exhibit significant differences in genetic background as evidenced by the distinct expression patterns of STs. Therefore, our findings may not be generalizable to other cell lines. Additionally, our exploratory analysis only involved nine primary samples, which may be insufficient to draw robust conclusions. The experiment was limited to cell culture model investigations, and further in vivo analyses are required to confirm the impact of glycation on the expression of STs and the underlying mechanisms. In the future, analysis using flow cytometry should be conducted to measure the overall increase in sialylation after glycation.

## Figures and Tables

**Figure 1 cells-12-02758-f001:**
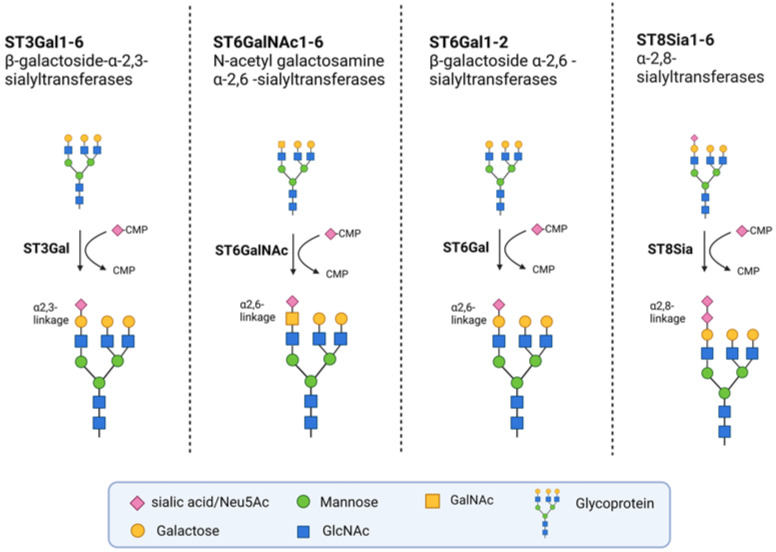
Groups of STs. The STs are grouped into four families according to the bonds they synthetize. They transfer activated CMP-Neu5Ac onto Galactose, N-Acetylgalactosamine (GALNAC), or Neu5Ac moieties of glycoproteins. The ST3Gal family members transfer sialic acids to terminal galactose residues via α-2,3 linkages, whereas the two known members of the ST6Gal family do so via α-2,6 linkages. The six members of the ST6GALNAC family transfer sialic acids from GALNAC residues via α-2,6-linkages. The members of the ST8SIA family transfer sialic acids to other terminal sialic acid residues via α-2,8-linkages. Created with Biorender.com (agreement number: NV255ORK3Q).

**Figure 2 cells-12-02758-f002:**
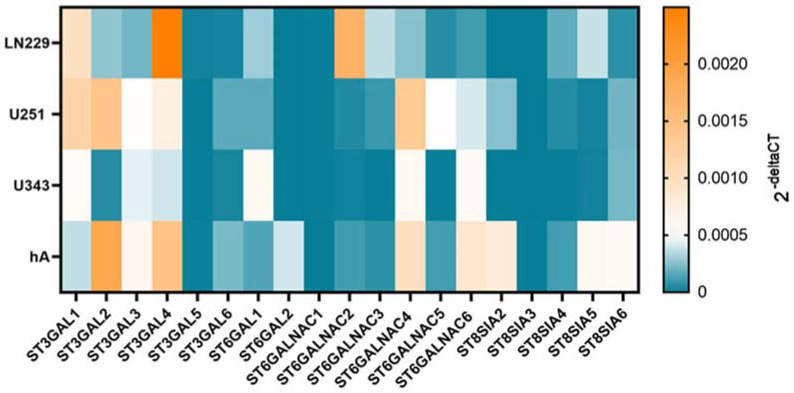
mRNA levels of STs in LN229, U251, U343, and hA. Heatmap of sialyltransferase mRNA levels in LN229 cells, U251 cells, U343 cells, and hA measured with qPCR.

**Figure 3 cells-12-02758-f003:**
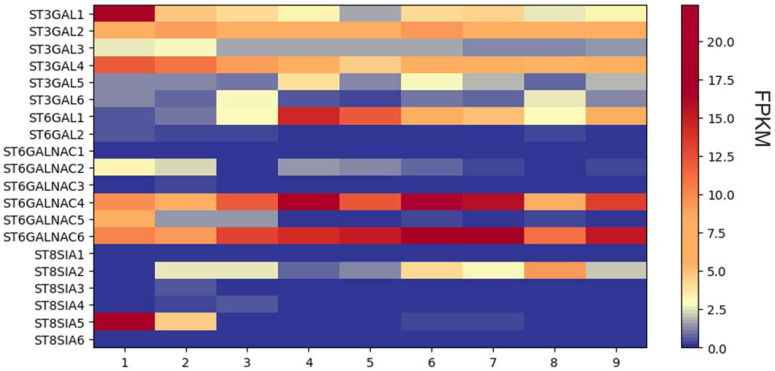
RNAseq analysis of sialyltransferase expression in nine primary glioma cells. Heat map of sialyltransferase expression levels. FPKM, fragments per kilobase per million fragments.

**Figure 4 cells-12-02758-f004:**
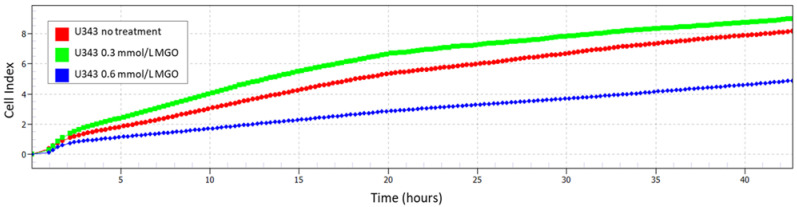
Cell index with and without MGO treatment. Impedance was measured with xCELLigence RTCA eSight. The graph shows a representative real-time analysis of U343 cells after treatment with 0.3 mmol/L MGO (green line), 0.6 mmol/L MGO (blue line), and without MGO (red line) over 42 h.

**Figure 5 cells-12-02758-f005:**
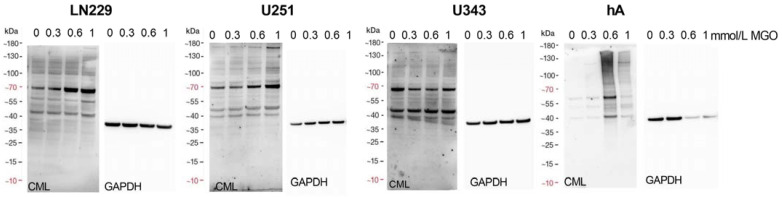
Glycation of glioma cell lines and hA. Immunoblot of LN229 cells, U251 cells, U343 cells, and hA with different MGO concentrations (**left**). Antibodies against carboxymethyl lysine (CML) were used to detect glycation. GAPDH was used as loading control (**right**).

**Figure 6 cells-12-02758-f006:**
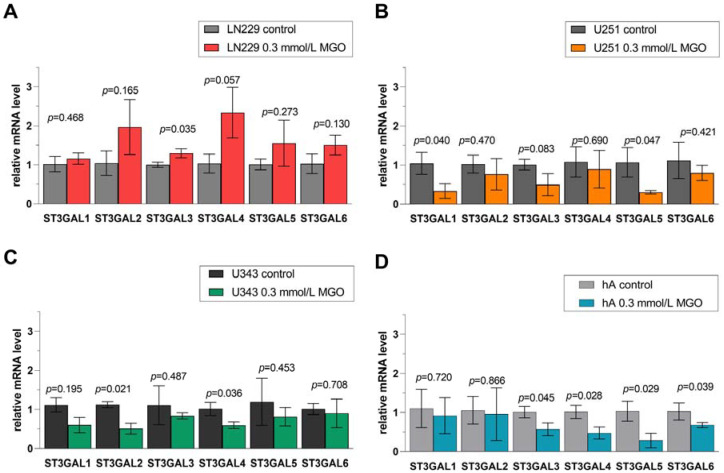
Relative mRNA levels of *ST3GAL1-6* in LN229 cells (**A**), U251 cells (**B**), U343 cells (**C**), and hA (**D**) measured with qPCR. Graphs show mRNA levels after 24 h with 0.3 mmol/L MGO normalized to the mean value of control cells. Student’s *t*-test was performed for statistical analysis. Error bars represent the means and SD of four independent biological replicates.

**Figure 7 cells-12-02758-f007:**
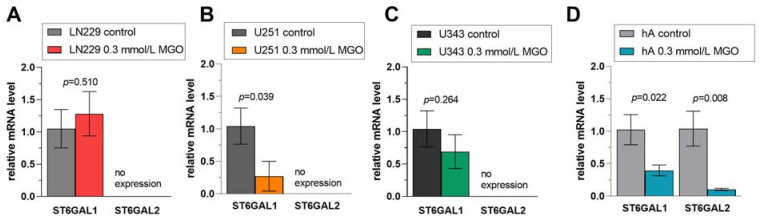
Relative mRNA levels of *ST6GAL1-2* in LN229 cells (**A**), U251 cells (**B**), U343 cells (**C**), and hA (**D**) measured with qPCR. Graphs show mRNA levels after 24 h with 0.3 mmol/L MGO. Student’s *t*-test was performed for statistical analysis. Error bars represent the means and SD of four independent biological replicates.

**Figure 8 cells-12-02758-f008:**
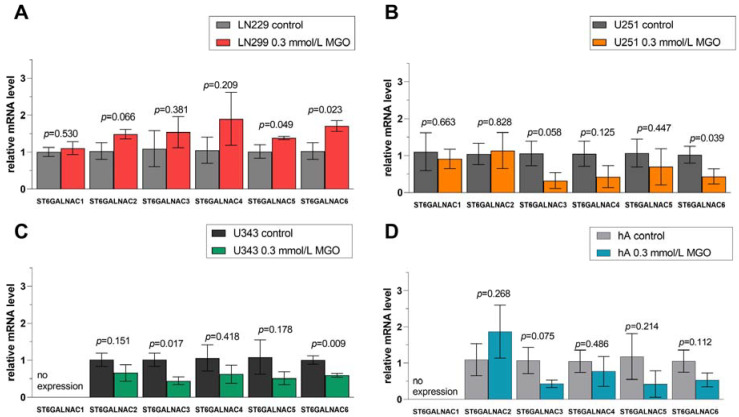
Relative mRNA levels of *ST6GALNAC1-6* in LN229 cells (**A**), U251 cells (**B**), U343 cells (**C**), and hA were measured with qPCR (**D**). Graphs show mRNA levels after 24 h with 0.3 mmol/L MGO normalized to the mean value of control cells. Student’s *t*-test was performed for statistical analysis. Error bars represent the means and SD of four independent biological replicates.

**Figure 9 cells-12-02758-f009:**
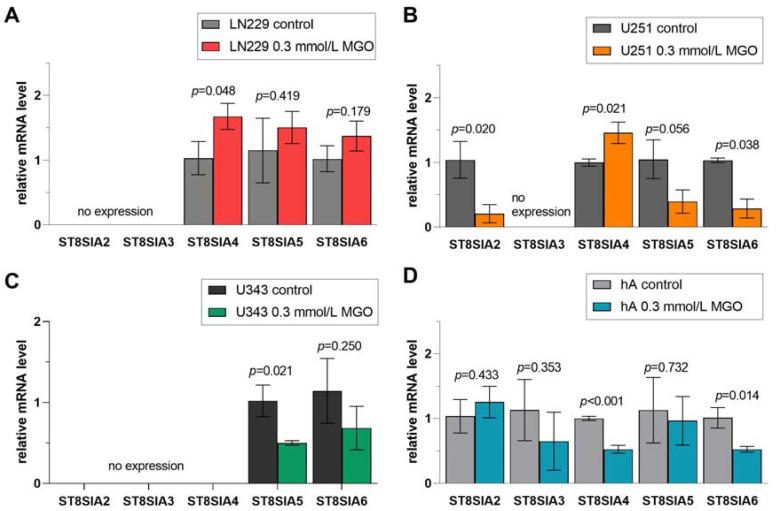
Relative mRNA levels of *ST8SIA1-6* in LN229 cells (**A**), U251 cells (**B**), U343 cells (**C**), and hA (**D**) measured with qPCR. Graphs show mRNA levels after 24 h with 0.3 mmol/L MGO. Student’s *t*-test was performed for statistical analysis. Error bars represent the means and SD of four independent biological replicates.

**Figure 10 cells-12-02758-f010:**
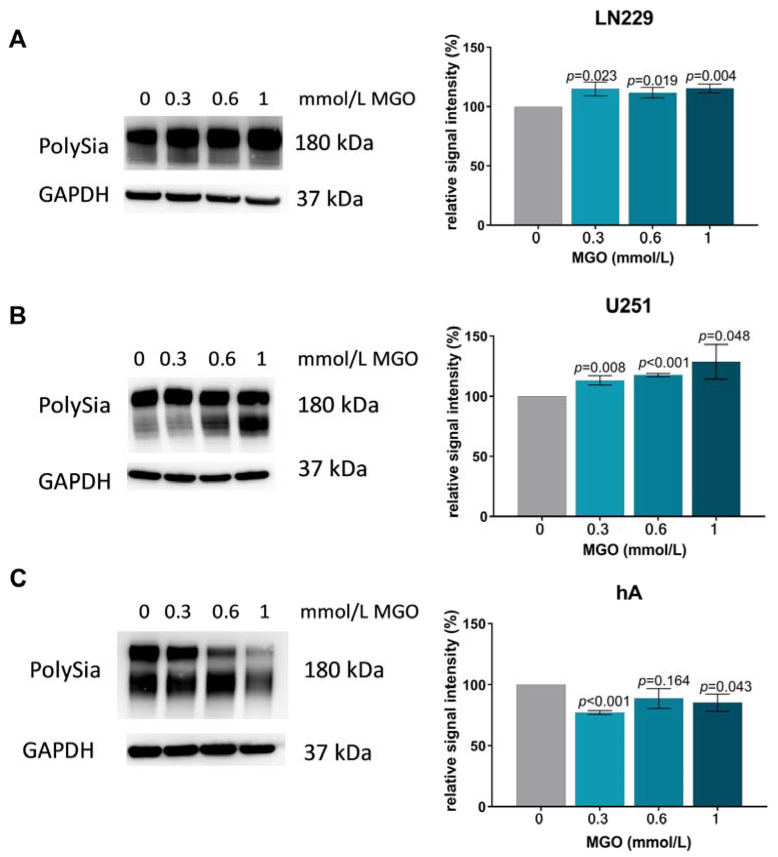
Polysialylation after MGO treatment. Immunoblot of LN229 cells (**A**), U251 cells (**B**), and hA (**C**) with different MGO concentrations (0, 0.3, 0.6, and 1 mmol/L) (left column). Polysialylation was detected with PolySia antibody. Graphs (right column) show representative quantification of the blot, normalized to the untreated cells. GAPDH was used as loading control. Student´s *t*-test was performed for statistical analysis. Graphs represent the means and SD of three independent biological replicates.

**Figure 11 cells-12-02758-f011:**
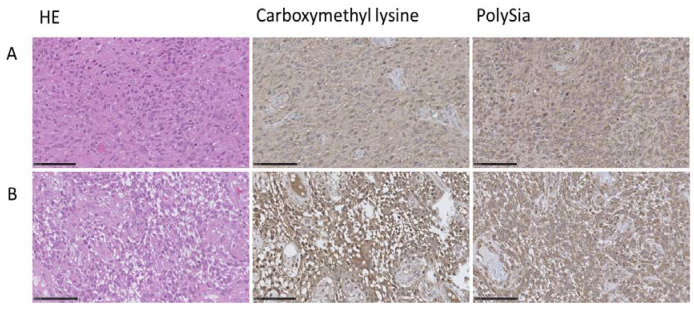
IHC analysis of carboxymethyl lysine and PolySia in GBM tissue. Representative IHC analysis including hematoxylin (HE), carboxymethyl lysine, and PolySia staining in different GBM tissue (**A**,**B**). Bar = 100 µm.

**Table 1 cells-12-02758-t001:** Antibodies used for Western blotting.

Antibody	Species	Dilution	Dilution Buffer	Manufacture
Monoclonal 735 anti-polySia antibody	*Mouse* IgG	1:2000	5% MP in TBS-T	Kind gift from Rita Gerardy-Schahn, Hannover Medical School, Germany), self-made
GAPDH (14C10)	*Rabbit* IgG	1:1000	5% BSA in TBS-T	Cell Signaling Technology Inc. (Danvers, MA, USA)
Anti-rabbit IgG, HRP-linked Antibody	*Goat*	1:1000	2% MP in TBS-T	Cell Signaling Technology Inc. (Danvers, MA, USA)
Anti-mouse IgG, HRP-linked Antibody	*Horse*	1:1000	2% MP in TBS-T	Cell Signaling Technology Inc. (Danvers, MA, USA)
Anti-Carboxymethyl Lysine antibody (ab125145)	*Mouse* IgG	1:1000	5% MP in TBS-T	Abcam (Cambridge, UK)

Abbreviations: BSA, bovine serum albumin; IgG, immunoglobulin G; MP, milk powder.

**Table 2 cells-12-02758-t002:** Primers used for real-time quantitative PCR.

Gene Name	Oligo Sequence 5’ to 3’ (Forward, Reverse)	Product Length (bp)	Reference Sequence
*ST3GAL1*	ATGTTGGGACCAAGACCACC, ACAAGTCCACCTCATCGCAG	321	NM_173344.3
*ST3GAL2*	CAGATAGTGCCTGGCGAGAA, CACTGGGGCGTAGGTGAATC	333	NM_006927.4
*ST3GAL3*	CCTTTCGCAAGTGGGCTAGA, AGAGAATCGCGCTCGTACTG	337	NM_001270459.2
*ST3GAL4*	CCTACAACAAGAAGCAGACCATTC, CTGGATCTCGGCTCCATAAGAG	335	NM_006278.3
*ST3GAL5*	ATGCGGACGAAGGCGG, ACAAGCTGGGCCTTCTCATC	187	NM_003896.4
*ST3GAL6*	GACCTCAAGAGTCCTTTGCAC, TTCACAGAAATTAAGCTGGTGGTT	297	NM_001323362.2
* ST6GAL1 *	TCCCAAAGTGGTACCAGAATCC, CTTCTCATAGAGCAGCGGGT	332	NM_003032.3
* ST6GAL2 *	TCTGCTCCTACACGTGGTTATG, AGAAGATGGTGGGTTTGGTTGA	336	XM_047446026.1
*ST6GALNAC2*	GCACGCCTATTTTGGACCAG, TCCAGGGACAGATCGTGGTT	291	NM_018414.5
*ST6GALNAC3*	TGGCCTGCATCCTGAAGAGAA, CTTTGGTGGGGGCATTGTTC	339	XM_047417095.1
* ST6GALNAC4 *	CGTGGTCTATGGGATGGTCAG, TGGAGTGTGATGGCTTGGGA	229	NM_175040.4
*ST6GALNAC5*	GGTCTGGCAGTGTGTTTAGC, TGCATTTTCAGGGGCTTGTG	254	NM_030965.3
* ST6GALNAC6 *	CGGTCAGCAGTGTTCGTGA, GCGGTAGGTGGTCTTGTTGC	339	NM_001388489.1
* ST8SIA2 *	CAGAGATCGAAGAAGAAATCGGGA, TGGGACACATTCATGGTGCT	334	NM_006011.4
* ST8SIA3 *	ATTTGGCGCTTTCCGTTTGG, GCAACATGTCAACAGGTACTGG	323	NM_015879.3
* ST8SIA4 *	CTCCTGTGGTGGAGTTTGCT, ACCTGTGCTGGGTCTTTTGAT	329	NM_005668.6
* ST8SIA5 *	AGTCTACTCTGTCCAGGTGCT, ACAGTGACCACATCCGTCTTC	328	NM_001307987.2
* ST8SIA6 *	GTAACCTACCCCCAACCACAG, TCATCAAGCCGGTGGACAAG	342	XM_024447978.2
*GAPDH*	TCGTGGAAGGACTCATGACC, TTCCCGTTCAGCTCAGGGAT	172	NM_002046.7

All primers are specific for *Homo sapiens* genes and have an annealing temperature of 60 °C.

**Table 3 cells-12-02758-t003:** Baseline data of glioma patients from which primary cultures were obtained.

Sample	Tumor	WHO Grade	IDH	MGMT	Age at Surgery	Sex
1	Glioblastoma	4	WT	unmethylated	64	m
2	Glioblastoma	4	WT	methylated	68	f
3	Glioblastoma	4	WT	methylated	73	f
4	Glioblastoma	4	WT	unmethylated	40	f
5	Oligodendroglioma	2	Mut	methylated	67	f
6	Oligodendroglioma	2	Mut	methylated	34	m
7	Astrocytoma	2	Mut	methylated	32	m
8	Astrocytoma	2	Mut	methylated	38	f
9	Ganglioglioma	1	WT	methylated	74	m

Abbreviations: IDH: isocitrate dehydrogenase; Mut: mutation, WT: wild-type; MGMT: O-(6)-methylguanine-DNA methyltransferase; f: female; m: male.

## Data Availability

The dataset is available from the corresponding author upon reasonable request.

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
