# Peer review of "Glycation Interferes with the Expression of Sialyltransferases and Leads to Increased Polysialylation in Glioblastoma Cells"

_cells, 2023, doi:10.3390/cells12232758_

Round 1

Reviewer 1 Report

Comments and Suggestions for Authors

The manuscript entitled “Glycation interferes with the expression of sialyltransferases and leads to increased polysialylation in glioblastoma cells” by Paola Schildauer and colleagues is an original research article about the in vitro evaluation of the glycation effect on sialyltransferase (ST) expression and polysialic acid (PSA) levels in different glioblastoma (GBM) cell lines, compared to human astrocytes (hA). Glycation represents a consequence of aerobic glycolysis (Warburg effect) carried out by GBM cells for energy production. The main product of glycation is methylglyoxal (MGO) a reactive dicarbonyl compound that causes protein alterations and cellular dysfunction. The authors examined the mRNA expression of different sialyltransferase families in three GBM cell lines (U251, LN229 and U343) compared to hA, and in nine primary glioma cells. They observed that in malignant cell lines ST mRNA levels were lower in comparison to hA with only few exceptions. The RNA sequencing analysis of STs performed on patient-derived glioma primary cultures confirmed this variability. The authors exposed GBM cell lines to various concentrations of MGO (0.3, 0.6, 1 mmol/L) for 24h and observed an increase of glycation confirming the suitability of MGO as glycation agent. In parallel, they treated GBM cell lines with MGO (0.3 mmol/ L) and observed that this treatment affects the mRNA expression of STs. Therefore, the authors examined the impact of glycation on polysialylation, they treated GBM cell lines with MGO (0.3 mmol/L) and analysed PSA levels, observing an increase of polysialylation in LN229 and U251 cell lines, in contrast to this, glycation decreased polysialylation in hA. Finally, the authors performed an immunohistochemical analysis in primary GBM tissues and observed an increase of PSA and carboxymethyl lysine (used to detect glycation).

The topic is up to date and is interesting to a general audience. Additionally, the manuscript is well written, and the English language is fine with only very few errors to correct throughout the text.

However, there are some points the authors should address before the manuscript could be accepted.

One major concern of the entire work is the classification of high-grade glioma. Authors are invited to better indicate Glioblastoma in accordance with the latest classification (Glioblastoma, IDH-wild type – CNS WHO grade 4).

Why do the authors included U343 cells (WHO grade 3) in their experiments? Some discordant results should be referred to this? Does the genetic background could be an issue for the discordant results? Please, discuss.

 The authors should pay more attention regarding the title of the research, which is “Glycation interferes with the expression of sialyltransferases and leads to increased polysialylation in glioblastoma cells”. The authors show big panels and give a general idea of the expression of all the sialyltranferases genes. This could be fine. Then, the authors focused mainly on the only less expressed genes that give rise to the elongation and synthesis of polysialic acid, namely ST8Sia2 and 4, which are slightly modulated and are not the only STs regulated by MGO treatment. If the goal of the manuscript was to demonstrate a direct effect of glycation on polysialylation, I suggest the authors to be more straight with their title that I would modify with: ‘’Glycation interferes with the expression of polysialyltransferases and leads to increased polysialylation in glioblastoma cells’’. Then, the authors are suggested to give more importance and show only the most relevant results about ST8Sia2 and 4 and Polysialic acid expression.

Based on what was previously written, I recommend the authors to carrying out experiment for lectin binding/lectin staining (with SNA and MAL-II) to analyze the amount of sialic acids. I suggest analyzing the binding through flow cytometry analysis.

One other major concern of the entire work is the classification of high-grade glioma. Authors are invited to better indicate Glioblastoma in accordance with the latest classification (Glioblastoma, IDH-wild type – CNS WHO grade 4).

Page 4, line 164, please correct “butter” with “buffer”.

Page 5, Table 1. Antibody used to detect glycation (anti-carboxymethyl lysine) was not included.

Authors also write that the antibody that recognize NCAM-Polysialic acid was a gift. I understand the situation but it is important to know if the origin of the antibody is homemade or if it is commercial and who is the manufacturer.

The authors conduct their experiments in three different GBM cell lines, U343, LN229 and U251, but analyze the MGO-dependent cell growth only in U343, the only cell line that is not affected by the mechanism they describe in the whole muanuscript. Is there a reason for this? I suggest repeating this experiment for the other two GBM lines and hA.

Page 8, in my opinion the paragraph 3.1 is very tedious for the reader, I recommend emphasizing only the significant results.

Page 10, line 335. Please, remove the term “lines” from the caption.

Page 12, figure 5A. Please, add the quantification of the various Western Blot analyses because they do not seem to support the described results. As you can see, GAPDH signals seem to change with the chosen treatment. I suggest the authors to use another housekeeping as beta-actin, beta-tubulin, alpha-tubulin that do not change with metabolic treatments and I think they will observe the expected results (please see Lee HJ, Howell SK, Sanford RJ, Beisswenger PJ. Methylglyoxal can modify GAPDH activity and structure. Ann N Y Acad Sci. 2005 Jun;1043:135-45. doi: 10.1196/annals.1333.017. PMID: 16037232).

Page 12, figure 5B, this figure summarizes the other graphs that are insert in subsequent paragraphs. In my opinion, the text is very repetitive in this way, so I recommend eliminating this figure.

Page 13, 14, 15, figure 6 A, B, C, D, figure 7 A, B, C, D, figure 8 A, B, C, D, figure 9 A, B, C, D.  Please remove the p value from the graphs and add an * to represent the significance of the results.

Page 15, line 440, please insert a space between “p=0.023” “and”.

Page 15, line 447-448, the authors write “Polysialylation was not detected in the U343            cell line and was not induced after MGO treatment”. The figure about this result is not present, please, show it.

Page 16, figure 10 B, C. What band was considered to make the quantification? It is not clear. In the Figure 10 C the intensity of the bands decreases as the MGO concentration increases, but the quantification graph represents a different result. Pay attention please. Polysialic acid is mainly attached to Neural Cell Adhesion Molecule (NCAM), I suggest that the authors evaluate by Western Blot the expression of NCAM in GBM cell lines to evaluate the degree of polysialylation.

Page 16, line 451, please, correct the MGO concentrations in the caption.

Page 16, paragraph 3.11. Do the patients chosen to do the IHC analysis fall among those described in Table 3? It is not clear which samples they are.

Page 20, line 616, please correct “Rose” with “Rosa”.

I believe the manuscript could be improved after this revision process before acceptance.

Reviewer 2 Report

Comments and Suggestions for Authors

The authors have analysed the effect of MGO on gene expression levels of ST genes in GBM across some cell lines suggesting a role for some specific genes in glycobiology.

While the technical execution seems sound, this is highly incremental work on a previous publication (https://www.mdpi.com/2073-4409/12/9/1219). Can the authors comment on the overlap and the advances made compared to the previous work? Apart from some expression profiling, which highlights high heterogeneity among the cell lines of ST genes, the molecular mechanism remain rather uncertain, with only one cell line is really diverting from the normal astrocytes. 

I am not convinced that the stains of the human GBM samples are specific and seem to stain everything, including infiltrating immune cells, matrix, neuropil, etc. How does this observation connect to the observations in the cell lines?

Reviewer 3 Report

Comments and Suggestions for Authors

The authors examined mRNA expression of several sialyltransferases (STs) in nine primary glioma cell cultures, glioma cell lines, and in non-cancerous human astrocytes, as well as an effect of glycation on aforementioned mRNA levels. They also compared mRNA levels of their genes of interest (several STs) in their cell lines with publicly available microarray data from CNS tumors and normal tissue. They demonstrated that effect of glycation on ST expression is cell-line specific  

Line 20. The authors state “MGO, a reactive dicarbonyl compound causes protein alterations”. Proteins are not the only targets of MGO involved in cancerogenesis, please discuss in detail.

 Line 25. Why did the authors choose these particular cell lines (ST8SIA4 in the LN229 and U251) for their study? Please justify the choise and provide more info about their genetic background, if relevant.

Line 138. The authors state “Human Astrocytes (hA), obtained from Sciencell Research were used as a healthy cell line”. I suggest replacing it with “as a non-cancerous cell line”

Please discuss in details the difference between Glycation and Glycosilation, and its impact on cancer cells, cancer stem cells, anti-tumor immune response, tumor microenvironment

Line 352 The authors state “MGO (0.3 mmol/L) stimulated cell proliferation (Supplemental Video 2, Figure 4), while higher concentration of MGO treatment (0.6 mmol/L) led to reduced cell proliferation”. How do you explain this observation? Is it just a non-specific cytotoxicity?

Why MGO was chosen as a glycation agent, but not glucose/ribose etc?

Line 378. “Antibodies against carboxymethyl lysine (CML) were used to detect glycation. GAPDH was used as a loading control” I wonder why the authors did not observe any changes in the levels of GAPDH LN229 glycation (I assume it was glycated too).

In figure 5, staining with GAPDH, U251, U343 – it looks like part of image was removed next to the GAPDH staining (there is an area what looks like a “cut through line” under the GAPDH line. Please reassure me that the images are original unmodified ones.  Perhaps provide additional images from independent western blotting experiment.

Line 514. This sentence sounds odd “In GBM, patients with the mesenchymal subtype Chong et al. found that ST3GAL1 is activated by the TGFβ signaling pathway and controls brain tumor formation”

Overall, unless my concerns regarding figure 5 are not addressed, I find that this manuscript should undergo significant revisions.

Round 2

Reviewer 3 Report

Comments and Suggestions for Authors

All my comments have been addressed.

Author Response

We would like to thank the reviewer for his time and effort.